# Oxidation-stable amine-containing adsorbents for carbon dioxide capture

Kyungmin Min[1], Woosung Choi[1], Chaehoon Kim[1] & Minkee Choi[1]

Amine-containing solids have been investigated as promising adsorbents for $CO_2$ capture, but the low oxidative stability of amines has been the biggest hurdle for their practical applications. Here, we developed an extra-stable adsorbent by combining two strategies. First, poly(ethyleneimine) (PEI) was functionalized with 1,2-epoxybutane, which generates tethered 2-hydroxybutyl groups. Second, chelators were pre-supported onto a silica support to poison p.p.m.-level metal impurities (Fe and Cu) that catalyse amine oxidation. The combination of these strategies led to remarkable synergy, and the resultant adsorbent showed a minor loss of $CO_2$ working capacity (8.5%) even after 30 days aging in $O_2$-containing flue gas at 110 °C. This corresponds to a ~50 times slower deactivation rate than a conventional PEI/silica, which shows a complete loss of $CO_2$ uptake capacity after the same treatment. The unprecedentedly high oxidative stability may represent an important break-through for the commercial implementation of these adsorbents.

[1] Department of Chemical and Biomolecular Engineering, Korea Advanced Institute of Science and Technology (KAIST), Daejeon 34141, Republic of Korea. Kyungmin Min and Woosung Choi contributed equally to this work.  Correspondence and requests for materials should be addressed to M.C. (email: mkchoi@kaist.ac.kr)

Carbon capture and storage (CCS) has been investigated as an important option to reduce anthropogenic $CO_2$ emissions[1,2]. $CO_2$ adsorption using aqueous amine solutions (e.g., monoethanolamine) is considered a benchmark technology for postcombustion $CO_2$ capture[3]. However, despite the several decades of optimization, the technology still has inherent limitations including volatile amine loss, reactor corrosion, and the high energy consumption for regeneration[1,4,5]. To overcome these limitations, solid adsorbents that are noncorrosive and can lower the energy consumption have emerged as potential alternatives[1,6–8]. Among various adsorbents, amine-containing solids are considered to be the most promising adsorbents because of high $CO_2$ adsorption selectivity in a typical flue gas containing dilute $CO_2$ (<15% $CO_2$)[6–8]. Such materials can be prepared by the heterogenization of amines in porous supports via the impregnation of polymeric amines such as poly(ethyleneimine) (PEI)[9–29], the grafting of aminosilanes[14,20,23,24,29–39], or the polymerization of amine monomers within the support pores[14,20,39,40].

For commercial implementation of adsorbents, they should be stable upon repeated $CO_2$ adsorption–desorption cycles over a long period. Generally, the low adsorbent stability necessitates the continuous addition of fresh adsorbents, which significantly increases the material cost for $CO_2$ capture. Amine-containing adsorbents are known to degrade via various chemical pathways including the oxidative degradation of amines[15–22,30–33], urea formation under the $CO_2$-rich atmosphere[21–25,33–35], steam-induced degradation of the porous supports[20,26–28], and irreversible adsorption of $SO_2$[29,37,38]. Aside from the oxidative degradation of amines, various solutions have been proposed for suppressing the degradation pathways. For instance, urea formation can be inhibited by selectively using secondary amines rather than primary amines[24,34] or injecting steam during adsorbent regeneration[14,22,23,28]. Steam-induced degradation can be solved by using porous supports with enhanced hydrothermal stability[26–28]. In addition, the amine poisoning by $SO_2$ can be avoided by employing advanced desulfurization unit before $CO_2$ capture[41]. Therefore, the oxidative degradation of amines remains the biggest hurdle for the practical applications of these adsorbents.

The most ubiquitous impurity in flue gas is $O_2$ (3–4%)[1]. Unfortunately, in the presence of $O_2$, the amine-containing adsorbents undergo rapid deactivation because of amine oxidation[15–22,30–33]. Amine oxidation is known to proceed via free radical formation by the reaction of $O_2$ with amines at elevated temperatures[42–44]. The oxidation rate significantly depends on the amine structures. The isolated primary amines are known to be more stable than the isolated secondary amines[30,31]. In the case of PEI, the linear PEI mainly containing secondary amines is more stable than the branched PEI with a mixture of primary, secondary, and tertiary amines[16]. The results indicated that the co-existence of different types of amines can affect the oxidative degradation of amines. The polymers with only distant primary amines such as poly(allylamine) are more stable than conventional PEI[17]. Recently, the use of propylene spacers between amine groups has been found to substantially increase amine stability compared with the PEI containing ethylene spacers[18]. It has also been reported that the addition of poly(ethylene glycol) into polymeric amines can retard the amine oxidations due to hydrogen bonds between the hydroxy groups of poly(ethylene glycol) and the amines[15]. Despite these efforts, it still remains a great challenge to improve the oxidative stability of amines to a commercially meaningful level (i.e., stable over several months).

In the present work, we report the synthesis of a modified PEI/silica that shows unprecedentedly high oxidative stability. The adsorbent was prepared by combining two strategies. First, PEI was functionalized with 1,2-epoxybutane (EB), which generates tethered 2-hydroxybutyl groups. Second, small amounts of chelators (<2wt%) were pre-supported into a silica support before the impregnation of functionalized PEI. We discovered that the polymeric amines contain p.p.m.-level metal impurities including Fe and Cu, which catalyse amine oxidation. The addition of chelators as a catalyst poison could significantly suppress the rate of amine oxidation. Notably, the combination of two strategies resulted in great synergy, compared to when each method was used separately.

## Results

**PEI vs EB-functionalized PEI**. The EB-functionalization of PEI (Nippon Shokubai, Epomin SP-012, MW 1200) was carried out as reported previously[21] (Fig. 1). We reported that the functionalization can significantly increase the $CO_2$ adsorption kinetics and amine stabilities against urea formation, while it can also retard the amine oxidation to some extent[21]. 50 wt% of PEI and EB-functionalized PEI (EB-PEI) were supported onto a macroporous silica synthesized by spray-drying a water slurry containing a fumed silica[21,26]. The silica has a high $CO_2$ accessibility to the supported amines and excellent steam stability arising from its large pore diameter (56 nm in average) and thick framework (10–15 nm)[26]. The $CO_2$ adsorption–desorption behaviours of PEI/$SiO_2$ and EB-PEI/$SiO_2$ were compared under practical temperature swing adsorption (TSA) conditions; $CO_2$ adsorption was carried out with a simulated flue gas (15% $CO_2$, 10% $H_2O$, $N_2$

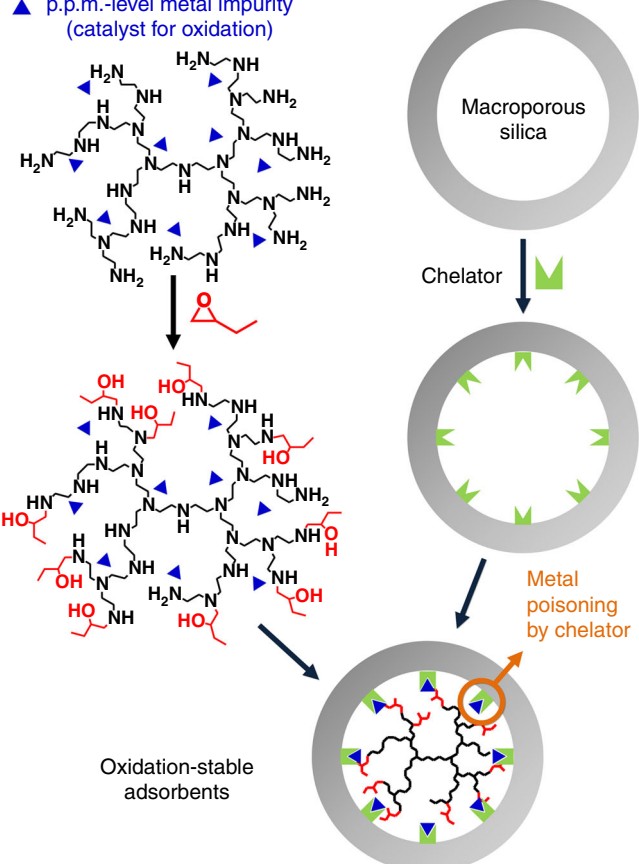

**Fig. 1** Synthesis of oxidation-stable $CO_2$ adsorbent. PEI was functionalized with 1,2-epoxybutane (EB), which generates tethered 2-hydroxybutyl groups. Small amounts of chelators were also pre-supported into a silica support to poison p.p.m.-level metal impurities in the polymeric amines that can catalyse the oxidative degradation of amines

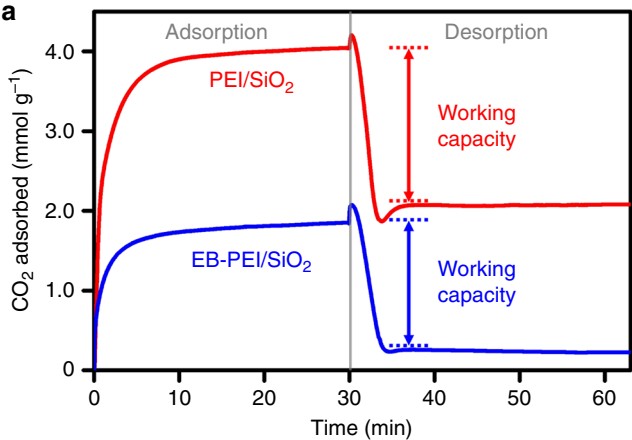

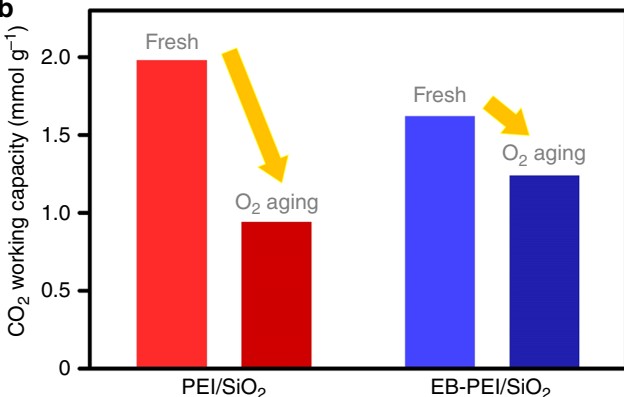

**Fig. 2** Comparison of the $CO_2$ adsorption–desorption behaviours and oxidative stabilities of PEI/SiO$_2$ and EB-PEI/SiO$_2$. **a** $CO_2$ adsorption–desorption profiles of the fresh adsorbents (adsorption: 15% $CO_2$, 10% $H_2O$, in $N_2$ balance at 60 °C; desorption: 100% $CO_2$ at 110 °C). **b** $CO_2$ working capacities before and after the oxidative aging under 3% $O_2$ in $N_2$ at 110 °C for 1 day

balance) at 60 °C and desorption was carried out under 100% $CO_2$ at 110 °C. As shown in Fig. 2a, EB-PEI/SiO$_2$ (1.84 mmol g$^{-1}$) showed smaller $CO_2$ adsorption than PEI/SiO$_2$ (4.05 mmol g$^{-1}$) due to the reduced amine density after the EB-functionalization of PEI[21]. However, the amounts of desorbable $CO_2$ during the adsorbent regeneration (i.e., $CO_2$ working capacities) were similar for both samples (1.98 and 1.62 mmol g$^{-1}$ for PEI/SiO$_2$ and EB-PEI/SiO$_2$, respectively), because of inefficient $CO_2$ desorption in PEI/SiO$_2$ at 110 °C. The more efficient regeneration of EB-PEI/SiO$_2$ can be attributed to the fact that EB-functionalization lowers the heat of $CO_2$ adsorption by generating tethered 2-hydroxybutyl groups on amines. The functional groups are electron-withdrawing groups that lower the amine basicity and also provide steric hindrance, lowering the heat of $CO_2$ adsorption[21]. According to differential scanning calorimetry, EB-PEI/SiO$_2$ exhibited a substantially lower heat of $CO_2$ adsorption (66.2 kJ mol$^{-1}$) than PEI/SiO$_2$ (80.5 kJ mol$^{-1}$).

The $CO_2$ working capacities of PEI/SiO$_2$ and EB-PEI/SiO$_2$ before and after oxidative aging under 3% $O_2$ in $N_2$ at 110 °C for 24 h are shown in Fig. 2b. The adsorbents were aged at the adsorbent regeneration temperature (110 °C), because it is the highest temperature the adsorbents can experience in TSA. PEI/SiO$_2$ showed a 52% decrease in $CO_2$ working capacity after aging, whereas EB-PEI/SiO$_2$ showed only a 23% decrease. These results show that the EB-functionalization of PEI can increase the oxidative stability of amines, which is consistent with our earlier

results[21]. The EB-functionalization converts the majority of the primary amines in PEI to secondary amines by alkylation with 2-hydroxybutyl groups. $^{13}$C NMR analysis showed that the primary:secondary:tertiary amine ratio in PEI is 36:37:27, whereas EB-PEI has a 10:56:34 ratio[21]. As Sayari pointed out, the oxidative degradation of amines in the PEI-type polymers is significantly affected by the co-existence of different types of amines[16]. Therefore, the increased stability of EB-PEI/SiO$_2$ might originate from the increased portion of secondary amines at the expense of primary amines. Alternatively, it can also be attributed to the generation of abundant hydroxy (−OH) groups after EB-functionalization, which can form hydrogen bonds with nearby amines. Chuang et al. reported that oxidative stability of amines could be improved in the presence of additives containing hydroxy groups (e.g., polyethylene glycol) because of their abilities to form hydrogen bonds with amines[15]. FT-IR spectra (Supplementary Fig. 1) showed that N-H stretching bands (3360 and 3290 cm$^{-1}$) became significantly broadened and the intensity of shoulder at 3160 cm$^{-1}$ increased after the EB-functionalization of PEI, which could be attributed to the presence of amines hydrogen-bonded with hydroxy groups[15].

**Poisoning of metal impurities in amines.** By coincidence, we discovered that the commercial PEI contains p.p.m.-level transition metal impurities, mainly Fe (17 p.p.m.) and Cu (6.9 p.p.m.) (Supplementary Table 1). Therefore, the EB-functionalized PEI had a more or less similar metal impurity content (Supplementary Table 1). Even though their concentrations are small, these metal species are known to catalyse amine oxidation by facilitating free radical formation via reaction with amines[42–44]. The $Fe^{3+}$/$Fe^{2+}$ and $Cu^{2+}$/$Cu^+$ redox cycles can oxidize amines to form amine radicals via single electron transfer[42–44], which can significantly increase the amine oxidation rate. The catalytic effects of these metal species on amine oxidation have been comprehensively investigated in the case of aqueous amine solutions, because Fe ions are continuously leached out by reactor corrosion and Cu ions are often deliberately added into the amine solutions to inhibit reactor corrosion[42–44]. However, in the case of solid adsorbents, possible contamination of commercial amine sources with these metal impurities and their catalytic effects on amine oxidation have been completely ignored.

To study the catalytic effects of the metal impurities on amine oxidation, we pre-supported six different types of chelators (Fig. 3a) as catalyst poisons onto a silica support. Then, the silicas were further impregnated with PEI and EB-PEI (Fig. 1). The ionic chelators were not soluble in the polymeric amines (PEI and EB-PEI) and thus needed to be pre-dispersed on the silica surface to maximize their interaction with metal impurities. Because we introduced minor amounts of chelators (<2 wt% per composite), the $CO_2$ working capacities of the adsorbents were not affected by the presence of the chelators (Supplementary Tables 2, 3). All chelators were effective in inhibiting amine oxidation (Fig. 3b, c). As the loading of chelators increased, more $CO_2$ working capacities were retained after oxidative aging. Among the various chelators, phosphate or phosphonate sodium salts (1–3 in Fig. 3a) were the most effective at inhibiting amine oxidation in both PEI (Fig. 3b) and EB-PEI (Fig. 3c) systems. Notably, trisodiumphosphate (TSP, 1 in Fig. 3a), which is a completely inorganic and very economic chelator, showed the most promising inhibition effect. This is practically important because organic chelators themselves are known to be oxidatively degraded in the presence of oxygen and metal species[45]. This is why organic chelators are not generally regarded as good oxidation inhibitors in aqueous amine solutions[44]. Interestingly, TSP is known as a weak chelator that is ineffective in poisoning metal impurities in aqueous amine

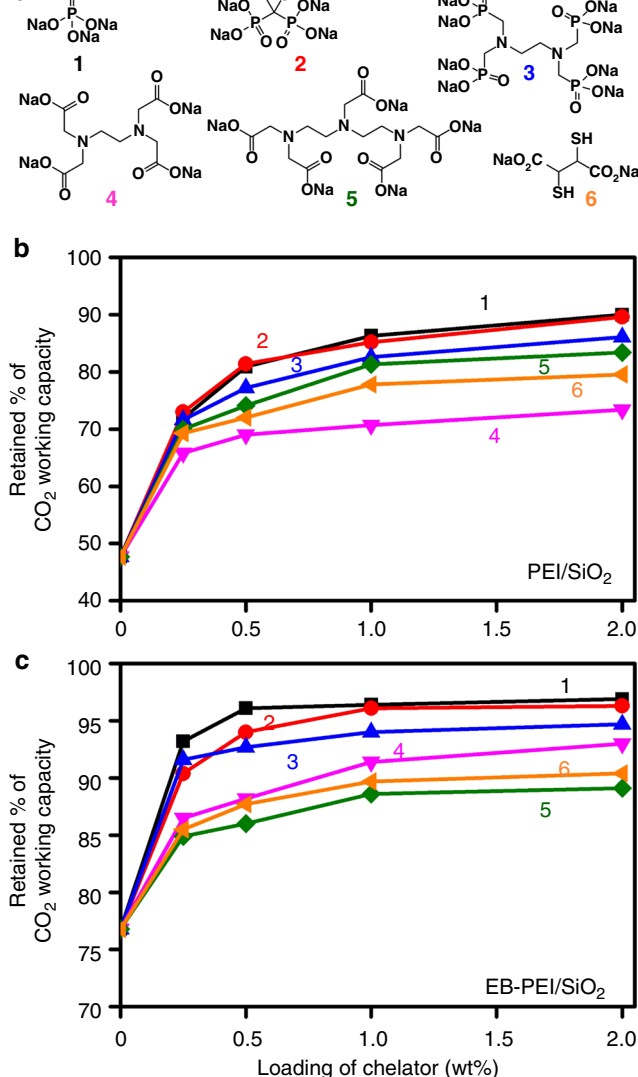

**Fig. 3** Poisoning of metal impurities using chelators and its retardation effects on amine degradation. **a** Types of chelators. **b**, **c** Retained $CO_2$ working capacity (%) of PEI/SiO$_2$ (**b**) and EB-PEI/SiO$_2$ (**c**) containing different amounts of chelators after oxidative aging under 3% $O_2$ in $N_2$ at 110 °C for 1 day. The numbers on plots indicate the types of chelators shown in **a**

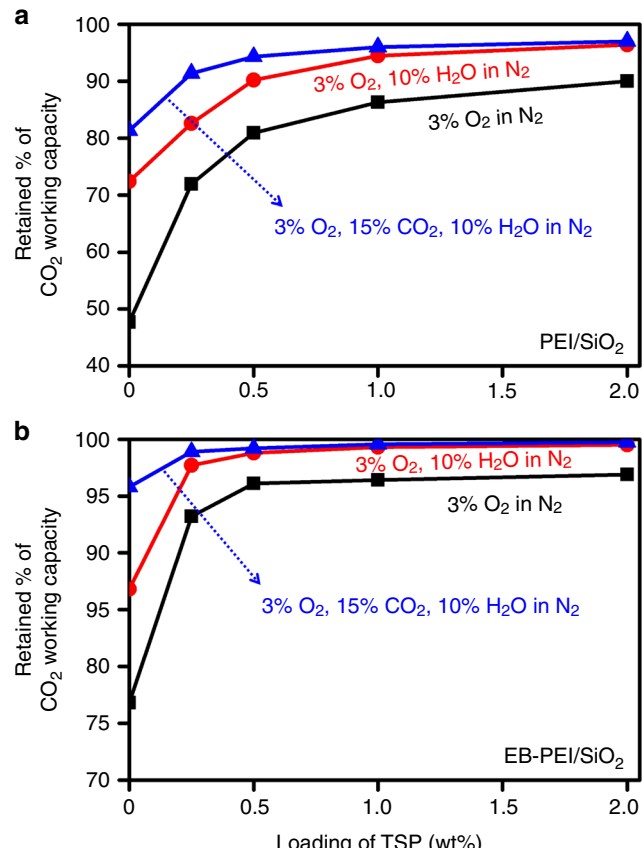

**Fig. 4** Effects of gas compositions on the oxidative degradation of adsorbents. **a**, **b** Retained $CO_2$ working capacity (%) of PEI/SiO$_2$ (**a**) and EB-PEI/SiO$_2$ (**b**) containing different contents of trisodiumphosphate (TSP), after oxidative aging under different gas compositions at 110 °C for 1 day

solutions[44]. We believe that even the TSP could efficiently poison the metal impurities in the solid adsorbents, because of the larger formation constant of corresponding metal phosphates in less polar polymeric amines (i.e., PEI and EB-PEI) than water. It has been reported that the formation constants of various metal complexes increase with decreasing solvent polarity[46].

**Long-term oxidative stabilities of adsorbents.** Because a real flue gas contains $CO_2$ and $H_2O$ in addition to $O_2$, the adsorbents must be stable under this atmosphere rather than under an $O_2/N_2$ mixed gas. Sayari thoughtfully pointed out that amine oxidation is slower in a real flue gas than in $O_2/N_2$ mixtures, because of the preferential interaction of amines with $CO_2$ and $H_2O$[22]. As shown in Fig. 4, we also confirmed that the addition of $CO_2$ and/or $H_2O$ can substantially retard amine oxidation in both of PEI/SiO$_2$ and EB-PEI/SiO$_2$ systems containing varied amounts of TSP. This means that the earlier aging experiments (Figs. 2 and 3) under 3% $O_2$ in $N_2$ should be considered as accelerated aging because they

were carried out in the absence of $CO_2$ and $H_2O$. Therefore, to evaluate the long-term stability in a more realistic condition, we analysed the $CO_2$ working capacities of adsorbents after aging in an $O_2$-containing flue gas (3% $O_2$, 15% $CO_2$, 10% $H_2O$ in $N_2$ balance) for 30 days.

To separately understand the effects of EB-functionalization and TSP addition on long-term oxidative stabilities, four adsorbents, i.e., PEI/SiO$_2$ and EB-PEI/SiO$_2$ prepared with and without 2 wt% TSP, were investigated (Fig. 5a). Fitting with the first-order deactivation kinetics (trend lines in Fig. 5a) showed that each of the EB-functionalization of PEI (EB-PEI/SiO$_2$) and the TSP addition into PEI (PEI/SiO$_2$ + 2 wt% TSP) could reduce the deactivation rate constant ($k_{deac}$) approximately to half (0.0546–0.0573 day$^{-1}$), compared to that of PEI/SiO$_2$ (0.123 day$^{-1}$). Notably, the combination of both strategies (EB-PEI/SiO$_2$ + 2 wt% TSP) led to remarkable synergy, and the $k_{deac}$ (0.00258 day$^{-1}$) became ~50 times smaller than that of PEI/SiO$_2$. The adsorbent retained 91.5% of the initial $CO_2$ working capacity after 30-days aging, while PEI/SiO$_2$ showed a complete loss of $CO_2$ capacity. The great synergy between the EB-functionalization and TSP addition was attributed to the lower polarity of EB-PEI than that of PEI due to the presence of ethyl side groups (Fig. 1), which can increase the formation constants of metal phosphates (i.e., stronger poisoning of metal species). Our adsorption experiments showed that EB-PEI/SiO$_2$ adsorbed much less $H_2O$ (0.7 wt%) than PEI/SiO$_2$ (4.7 wt%) under the simulated flue gas at 60 °C, which proved the lower polarity of EB-PEI.

In FT-IR spectra (Fig. 5b), PEI/SiO$_2$ without TSP showed a marked increase in C=O/C=N stretching bands (1670 cm$^{-1}$)

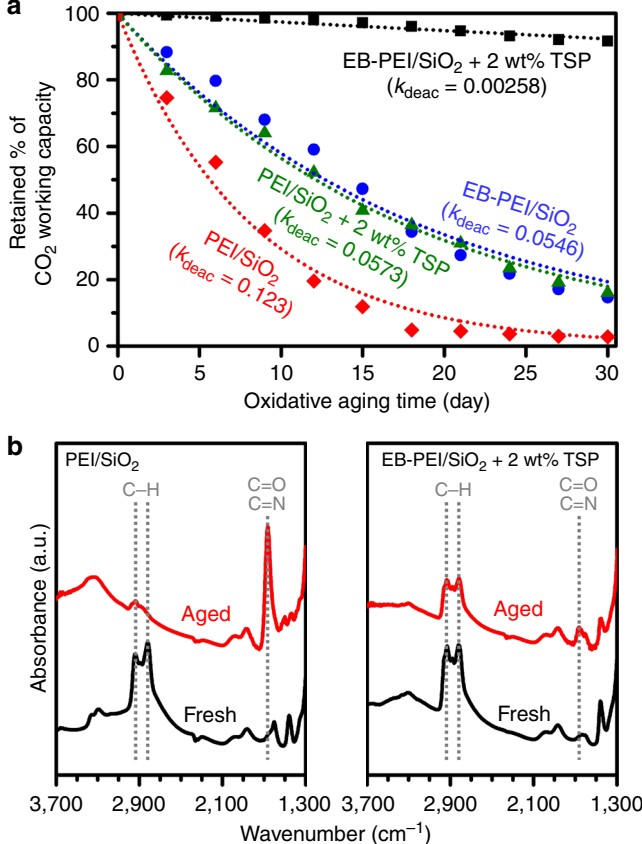

**Fig. 5** Long-term oxidative stabilities of adsorbents. **a** Retained $CO_2$ working capacity (%) of $PEI/SiO_2$ and $EB-PEI/SiO_2$ prepared with and without 2 wt% trisodiumphosphate (TSP) after aging under 3% $O_2$, 15% $CO_2$, 10% $H_2O$ in $N_2$ balance at 110 °C. The trend lines are obtained by fitting with the first-order deactivation model. **b** FT-IR spectra of $PEI/SiO_2$ without TSP and $EB-PEI/SiO_2$ with 2 wt% TSP, before and after 30-days aging

and a decrease in C–H stretching band (2800–3000 cm$^{-1}$) after the 30-days aging. The result indicates the formation of amide/ imine species[15,18,19,22]. We also found that $PEI/SiO_2$ lost 60% of the initial nitrogen content after the oxidative aging (Supplementary Fig. 2), which could be attributed to the hydrolysis of terminal imines to produce $NH_3$[42,43]. Notably, carbon content also substantially decreased (32%), which indicated a substantial polymer loss from the adsorbent. This implies that oxidation can partly cleave the polymer backbone into smaller fragments, which can be thermally evaporated. The chain degradation may be the consequence of complex amine oxidation reactions. It has been proposed that the ethylenediamine unit of PEI can be decomposed by oxidation into formamide and hemi-aminal[47], which can subsequently be decomposed into formic acid, amines, and imines. Alternatively, the imines generated at the middle of the polymer backbone can be hydrolysed to produce amine and aldehydes, which can also result in chain cleavage. Based on these results, it can be concluded that the oxidation of polymeric amines can lead to the loss of $CO_2$ capacity for two reasons: (1) the conversion of basic amines into non-basic amide/imine species, and (2) the accelerated loss of polymeric amines by oxidative fragmentation. In contrast, $EB-PEI/SiO_2$ with 2 wt% TSP exhibited insubstantial change in FT-IR spectrum after aging. Furthermore, the decreases in nitrogen and carbon contents were insignificant (<6.2%).

## Discussion

We synthesized oxidation-stable amine-containing adsorbents via the functionalization of PEI with 1,2-epoxybutane (EB) and the poisoning of the p.p.m.-level metal impurities with chelators. The combination of two different strategies could make remarkable synergy, because chelators form metal complexes more efficiently (i.e., stronger poisoning) in less polar EB-functionalized PEI. We believe that the amine-containing solid adsorbent is beneficial in terms of oxidative stability compared to conventional aqueous amine solutions, because the aqueous amine solutions cause the continuous leaching of metal species by reactor corrosion and these metals act as catalysts for amine oxidation. This requires the continuous and stoichiometric addition of inhibitors such as oxygen/radical scavengers or chelators, and also the removal of their decomposition products[44]. In the case of solid adsorbents, the initial poisoning of the p.p.m.-level metal impurities in polymeric amines is sufficient to increase the oxidative stability over a long period because there is no continuous metal buildup from reactor corrosion. Furthermore, the polymeric amines less polar than water can also enable more efficient metal poisoning by chelators. The unprecedentedly high oxidative stability of amine-containing solid adsorbents may represent an important breakthrough toward their commercial implementation.

## Methods

**Functionalization of PEI with 1,2-epoxybutane (EB)**. EB-functionalization of a commercial PEI (Nippon Shokubai, Epomin SP-012, MW 1200) was carried out as we reported previously[21]. In a typical synthesis, 12 g of 1,2-epoxybutane (Sigma-Aldrich, 99%) was added dropwise to 60 g of a 33 wt% methanolic solution of PEI. The molar ratio between 1,2-epoxybutane and the nitrogen content in PEI (22 mmol$_N$ g$^{-1}$) was fixed at 0.37. The reaction was carried out at room temperature for 12 h with stirring. The resultant methanolic solution containing 44 wt% EB-PEI was directly used for impregnation into a silica support.

**Preparation of a macroporous silica support**. A highly macroporous silica support was prepared following the procedure we reported previously[21,26]. The silica was synthesized by spray-drying a water slurry containing 9.5 wt% fumed silica (OCI, KONASIL K-300) and 0.5 wt% silica sol (Aldrich, Ludox AS-30) as a binder. In a typical synthesis, 0.95 kg fumed silica, 0.05 kg silica sol, and 9 kg water were mixed and the resultant slurry was injected for spray-drying. The spray-drying was carried out using a spray dryer with a co-current drying configuration and a rotary atomizer (Zeustec ZSD-25). The injection rate was 30 cm$^3$ min$^{-1}$, and the rotating speed of the rotary atomizer was set to 4000 r.p.m. The air blowing inlet temperature was 210 °C and the outlet temperature was 150 °C. The resultant silica particles were calcined in dry air at 600 °C to sinter the fumed silica into a three-dimensional porous network.

**Impregnation of chelators into the silica support**. In the present work, chelators in a completely basic form (all acidic functional groups were titrated with Na$^+$) were impregnated into the macroporous silica support using their aqueous solutions (typically, 0.25–2.0 wt% chelator concentrations). For the preparation of the aqueous solutions, trisodiumphosphate (**1** in Fig. 3a, Sigma Aldrich, 96%) and ethylenediaminetetraacetic acid tetrasodium salt dihydrate (**4**, Sigma Aldrich, 99%) were directly dissolved in deionized water. In the cases of chelators in an acidic form, such as 1-hydroxyethane 1,1-diphosphonic acid monohydrate (**2**, Sigma Aldrich, 95%), ethylenediamine tetramethylene phosphonic acid (**3**, Tokyo Chemical Industry, 98%), diethylenetriaminepentaacetic acid (**5**, Sigma Aldrich, 99%), and meso-2,3-dimercaptosuccinic acid (**6**, Sigma Aldrich, 98%), controlled amounts of NaOH were dissolved together in deionized water to completely titrate acidic H$^+$ into Na$^+$. After the wet impregnation of the chelators into the silica support, the samples were dried in a vacuum oven at 100 °C for 12 h. The final loading of chelators was controlled in the range of 0–4 wt% with respect to the mass of silica support.

**Preparation of the $PEI/SiO_2$ and $EB-PEI/SiO_2$ adsorbents**. Methanolic solutions of PEI and EB-PEI were impregnated into the macroporous silica supports pre-impregnated with 0–4 wt% chelators. After impregnation, the samples were dried at 60 °C for 12 h in a vacuum oven to remove methanol completely. With respect to the mass of the final composite adsorbents, the polymer content was fixed to 50 wt % and the chelator content was controlled in the range of 0–2 wt%.

**Material characterization**. The heat of $CO_2$ adsorption was measured by differential scanning calorimetry (Setaram Instrumentation, Setsys Evolution). Before

the measurements, the samples were degassed at 100 °C for 1 h under $N_2$ flow (50 $cm^3 min^{-1}$). Then, the samples were cooled to 60 °C. Subsequently, the gas was switched to 15% $CO_2$ (50 $cm^3 min^{-1}$). The heat of adsorption was calculated through the integration of the heat flow curve. Fourier transform infrared (FT-IR) spectra were recorded using an FT-IR spectrometer (Thermo Nicolet NEXUS 470). Prior to the analysis, 3 mg of the adsorbents were grinded with 18 mg of the silica (macroporous silica support) as a diluent. The mixtures were pressed into a self-supporting wafer. Before the FT-IR measurements, each sample was degassed at 100 °C for 3 h under vacuum in an in situ infrared cell equipped with $CaF_2$ windows. FT-IR spectra were collected at room temperature. Elemental compositions (C, H, and N) of the adsorbents were analysed using a FLASH 2000 (Thermo Scientific) instrument. The contents of transition metal impurities (Fe, Cu, etc.) in PEI and EB-PEI were determined by inductively coupled plasma mass spectroscopy (ICP-MS) using an ICP-MS 7700S instrument (Agilent). In the case of EB-PEI, the metal concentrations were determined after the complete evaporation of the methanol solvent at 60 °C for 12 h under vacuum after its synthesis.

**$CO_2$ adsorption–desorption experiments**. $CO_2$ adsorption–desorption profiles were collected by a thermogravimetric analysis-mass spectrometry (TGA-MS) setup[21,35]. Prior to the measurements, all samples were degassed at 100 °C for 1 h under $N_2$ flow (50 $cm^3 min^{-1}$). $CO_2$ adsorption was carried out using a simulated wet flue gas containing 15% $CO_2$, 10% $H_2O$, and $N_2$ balance at 60 °C. After 30 min adsorption, the gas was switched to 100% $CO_2$ flow (50 $cm^3 min^{-1}$) and the temperature was increased to 110 °C (ramp: 20 °C $min^{-1}$). Then the temperature was maintained for 30 min for the desorption process. The adsorbed amount of $CO_2$ was calculated by subtracting the amount of adsorbed $H_2O$ (determined by mass spectrometry) from the total mass increase determined by TGA. To confirm the reliability of the TGA-MS results, the $CO_2$ uptake was also cross-checked with an automated chemisorption analyser (Micromeritics, Autochem II 2920) specially equipped with a cold trap (−10 °C) for $H_2O$ removal in front of the TCD detector[21,35].

**Data availability**. The data that support the findings of this study are available from the corresponding author upon request.

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

## Acknowledgements

This work was supported by Korea CCS R&D Center (KCRC) grant funded by the Korea government (Ministry of Science, ICT & Future Planning) (NRF-2014M1A8A1049256) and the Basic Science Research Program through the National Research Foundation of Korea (NRF-2017R1A2B2002346).

## Author contributions

M.C. conceived and designed this study. K.M., W.C., and C.K. performed material synthesis, characterizations, and $CO_2$ adsorption–desorption experiments. M.C. wrote the manuscript with assistance from K.M., W.C., and C.K. All authors discussed the results and commented on the manuscript.

## Additional information

**Competing interests:** The authors declare no competing financial interests.

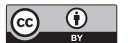

