## [Peer Review File · Nature Communications]

Reviewers' comments:

Reviewer #1 (Remarks to the Author):

In this work, authors combined several observations from the literature such as the inhibiting effect of CO₂ for oxidative degradation (Ref. 15), the catalytic effect of trace metals in the oxidation of amines in solution (Ref. 37-39), the inhibiting effect of chelating agents (Ref. 37-39) and the serendipitous discovery that commercial polyethylenimine (PEI) contains traces of metals to come up with an oxidation-resistant CO₂ amine-containing adsorbent.

Although I have a number of concerns to be detailed hereafter, I believe that this work would be fine after extensive improvement and data re-interpretation. Nonetheless, it does not rise in terms of novelty and significance, to the level of Nature Communications. It could be published in a more "regular" journal.

1. Although the PEI functionalization with epoxybutane is not the main topic of this work, I have some reservations about this procedure. First, using the optimum formulation selected by the authors, i.e. 037EB-PEI/SiO₂, the weight of the polymer increases by 62%. Therefore at 50% loading, the overall weight of the adsorbent increases by 31%! Moreover, the inactive tertiary amines increase by 22%, whereas the most active primary amines decrease by 56% (Ref. 17, Table 1). As a result, the amine efficiency (CO₂/N) whether based on all amines, or only on primary and secondary amines decreased [1.84 mmol/g (page 4) instead of the calculated 2.51 mmol/g if the efficiency (CO₂/N ratio) were preserved]. I am not sure that these shortcomings could be compensated by the advantages that functionalization imparts to the materials.
2. The large difference between the actual uptake before and after functionalization (4.05 vs. 1.84 mmol/g; page 2) was dismissed in favor of more comparable "working capacities" using pure CO₂ at high adsorption temperature (110 C) to remove CO₂ adsorbed at lower temperature (1.98 vs. 1.62 mmol/g; 18% decrease nonetheless). I understand that using pure CO₂ as a purge gas during desorption may allow the separation of high purity carbon dioxide, but because the small difference between adsorption and desorption temperatures, and the large difference in the partial pressure of CO₂ during adsorption (0.15 atm) and desorption (1 atm), make this procedure unattractive. Actually with the exception of this work and previous work by the same group (Ref. 16), I am not aware of any other paper using this approach. The use of steam seems to be more popular.
3. Some of the most insightful papers on oxidative degradation of supported amines (DOI: 10.1039/c3cp53928h; DOI: 10.1002/chem.201300864, and possibly DOI: 10.1021/ef4001067) were overlooked or ignored.
4. Some important statements are not correct (see also comment 5), for example in page 2, lines 44-45, authors reported (based on literature sources) that "urea formation can be inhibited by using secondary amines rather than primary adsorbent regeneration (12,15,26,35). I am not sure that this statement has been reported in (or can be drawn from) any the references cited. However, in reference 13 (Fig. 1), it was demonstrated that secondary amine does not form urea, but only when it is isolated. When it is not isolated, as in linear PEI, it does form cyclic urea, albeit the process is slower than in regular (branched) PEI.
5. Lines 101 to 105 "Notably, other studies reported that secondary amines are generally less stable than primary amines (29,30). We believe that the increased oxidative stability of EB-PEI/SiO₂, despite the increased secondary amine contents, can be attributed to the generation of abundant hydroxy (-OH) groups after EB-functionalization, which can form hydrogen bonding with nearby amines". It is correct that Ref. 29 and 30 reported that grafted propylamine (primary) is more air-resistant than grafted N-methyl propylamine (secondary). Nonetheless, this comparison is valid only for isolated amines. In polyethylenimines, the linear polymer (all secondary amines)

seems to be more resistant than the branched polymer (mixture of different amines), see DOI: 10.1039/c3cp53928h. Therefore, the increased resistance may be associated with the elimination of primary amines, and not necessarily associated with OH groups.

6. Supplementary Table 1. While the weight of PEI increased by 62% upon reaction with epoxybutane to obtain EB-PEI, the iron and copper content remained constant in both polymers (see also page 5, line 114). This is unrealistic. As for the other metals, although their concentrations are much smaller, their variations from PEI to EB-PEI are questionable.

7. The main problem of oxidative degradation may actually not be associated with the small amount of oxygen in the feed, since CO₂ has an inhibiting effect on oxidative degradation, and the adsorption is usually carried out at low temperature. The problem may be the cooling step (after desorption of CO₂) since the cheapest way would be to cool with air. That is another problem.

Reviewer #2 (Remarks to the Author):

This well-written manuscript presents studies that continue the authors' previous work in creating oxidatively-stable amine sorbents for CO₂ capture. Previously, the authors' work functionalized poly(ethylenimine) (PEI, a common aminopolymer used for CO₂ capture) with hydroxybutyl groups to improve the oxidative stability (ref. 17). This manuscript presents a simple strategy for modification of the oxide support to further improve the oxidative stability of these materials; this strategy is expected to be largely applicable to a variety of aminopolymer-based CO₂ sorbents and will be of interest to those working in the field. The manuscript should be published essentially as is, with only a couple of minor comments that the authors could consider.

Summary of key results:

The authors introduce the use of metal-chelating groups on the oxide support to remove trace amounts of metal impurity that come with commercially-available PEI. This strategy is combined with their previous PEI-functionalization strategy to produce a solid CO₂ sorbent with high CO₂ capacity and stability against oxidative degradation even after exposure to aggressive oxidizing environments for long time periods. Adsorption is performed from a simulated flue gas stream, and regeneration is performed in a CO₂ stream, thus generating a purified CO₂ product. The authors characterize the first-order degradation kinetics of their sorbents with the different treatments, demonstrating that the combination of PEI functionalization and introduction of metal-chelating groups has an additive effect on improving the oxidative stability.

Originality and significance:

The work is new in demonstrating that a simple and inexpensive modification to the solid oxide support, in addition to modifying the polymer that goes in that support, can have dramatic consequences in improving the long-term viability of solid sorbent materials for CO₂ capture.

Data and methodology:

The data are presented clearly, with the appropriate comparisons and control experiments.

Appropriate use of statistics and treatment of uncertainties:

The data do not appear to have any statistical treatment, and it is unclear whether experiments have been repeated. However, the trends in stability are clear as the loading of the metal chelators is increased, suggesting that the conclusions are qualitatively valid.

Conclusions:

The conclusions are supported by the data presented.

Suggested improvements:

It would be helpful if the authors could present a discussion of how to distinguish between the two

mechanisms of CO₂ capacity loss presented on page 7 (amine oxidation into amide/imine, and loss of amines by fragmentation). It is currently unclear how much of the capacity loss should be attributed to loss of amine content in the material, and how much to oxidation, though both appear to be suppressed by addition of the metal-chelating group. Additionally, it would be helpful if the authors could discuss how introduction of these metal chelators helps to suppress chain fragmentation.

Reviewer #3 (Remarks to the Author):

This paper reports that the amine-containing CO₂ adsorbents prepared via combining two strategies (functionalization of PEI with 1,2-epoxybutane and the poisoning of metal impurities with chelators) show the significantly enhanced oxidation stability even after 30 days-long aging in O₂-containing flue gas at 110 °C. I found this article certainly scientifically interesting, making me recommend its publication in Nature Communications. However, I have a couple of comments to be properly answered prior to its publication (see below).

1. The authors carried out the CO₂ adsorption-desorption experiments under practical temperature swing adsorption (TSA) conditions. CO₂ adsorption was performed using a wet flue gas containing 15% CO₂, 10% H₂O, and N₂ balance at 60 °C for 30 min and desorption was performed under 100% CO₂ at 110 °C for 30 min. I wonder how the authors treated the gas during decreasing temperature from 110 to 60 °C before moving to the next CO₂ adsorption step. This is very important because the remaining CO₂ gas or other carrier gas (i.e., He or Ar) can somehow influence the CO₂ working capacity of the solid adsorbents. In the case of zeolites X and A, the adsorbents can be partly regenerated under He or Ar flow even at atmospheric condition (25 °C and 1 bar). The author should clearly mention the whole experiment conditions.

2. In their previous study, the author stated that the amine-functionalized porous material may suffer from serious amine deactivation due to urea formation under the desorption conditions at temperatures higher than 120 °C. They indeed regenerated the samples under at 130 (Energy Environ. Sci., 2016, 9, 1803-1811) or 120 °C (Nat. Commun., 2016, 7), and successfully overcame the adsorbent deactivation. However, in the submitted ms, the adsorbents were aged at 110 °C only, which is lower by 10 °C than the amine deactivation temperature (120 °C) and the same as the pretreatment temperature (110 °C). I would like to know the reason the authors selected 110 °C as an ageing temperature.

Point-by-Point Response

We greatly appreciate all the reviewers' thoughtful comments and suggestions for improving the quality of our manuscript. It was clear that the reviewers carefully read the present manuscript, our earlier papers, and even related references. The comments were very constructive and important. We have prepared the following responses to fully address the reviewers' comments.

Reviewer #1

In this work, authors combined several observations from the literature such as the inhibiting effect of CO₂ for oxidative degradation (Ref. 15), the catalytic effect of trace metals in the oxidation of amines in solution (Ref. 37-39), the inhibiting effect of chelating agents (Ref. 37-39) and the serendipitous discovery that commercial polyethylenimine (PEI) contains traces of metals to come up with an oxidation-resistant CO₂ amine-containing adsorbent. Although I have a number of concerns to be detailed hereafter, I believe that this work would be fine after extensive improvement and data re-interpretation. Nonetheless, it does not rise in terms of novelty and significance, to the level of Nature Communications. It could be published in a more "regular" journal.

Point 1: Although the PEI functionalization with epoxybutane is not the main topic of this work, I have some reservations about this procedure. First, using the optimum formulation selected by the authors, i.e. 037EB-PEI/SiO₂, the weight of the polymer increases by 62%. Therefore at 50% loading, the overall weight of the adsorbent increases by 31%! Moreover, the inactive tertiary amines increase by 22%, whereas the most active primary amines decrease by 56% (Ref. 17, Table 1). As a result, the amine efficiency (CO₂/N) whether based on all amines, or only on primary and secondary amines decreased [1.84 mmol/g (page 4)] instead of the calculated 2.51 mmol/g if the efficiency (CO₂/N ratio) were preserved. I am not sure that these shortcomings could be compensated by the advantages that functionalization imparts to the materials.

Response 1: In the case of EB-PEI, polymer loading (50 wt%) was calculated based on the final polymer, not based on the original PEI. Other comments were reasonable. Nevertheless, we can clearly justify the significant benefits of PEI functionalization with 1,2-epoxybutane (EB). As the reviewer pointed out, the functionalization results in the reduction of amine content and thus CO₂ uptake. However, at the expense of CO₂ capacity, significant advantages in terms of chemical stability of amines can be achieved, as we reported in our earlier work [*Nat. Commun.* **7**, 12640 (2016), ref. 21 of the revised manuscript] and in the present work. More importantly, the ultimate metric for evaluating adsorbents is not the CO₂ working capacity, but the overall energy required for capturing CO₂ (i.e., mainly the heat required for adsorbent regeneration). However, most researchers, ourselves

included, have mainly focused on improving the CO₂ working capacity of adsorbents. According to our recent study (we are currently preparing a manuscript), the EB-PEI/SiO₂ requires much less heat than PEI/SiO₂ for capturing a fixed amount of CO₂. The heat required for adsorbent regeneration consists of three major components: *i*) sensible heat (the heat required for heating the adsorbents from adsorption to desorption temperature), *ii*) heat required for CO₂ desorption, and *iii*) heat required for H₂O desorption (adsorbents generally co-adsorb appreciable amounts of H₂O from a flue gas). For the estimation of heat requirements, we rigorously measured the specific heat capacities of adsorbents, the heat of adsorptions for CO₂ and H₂O, and the working capacities of CO₂ and H₂O. The resultant heat requirements are summarized below (Additional Fig. 1).

Additional Fig. 1 Regeneration heat required for the PEI/SiO₂ and EB-PEI/SiO₂ in the present TSA conditions.

The EB-PEI/SiO₂ showed substantially lower total heat requirement (2.66 GJ tCO₂⁻¹) than the PEI/SiO₂ (4.03 GJ tCO₂⁻¹). It is notable that the high CO₂ working capacity of PEI/SiO₂ only contributed to the reduction of sensible heat (the adsorbent with larger CO₂ capacity needs a smaller number of TSA cycles for capturing the fixed amount of CO₂). Instead, the strong interaction between PEI and CO₂ (*i.e.*, large heat of CO₂ adsorption) led to large heat requirement for CO₂ desorption. It is noteworthy that the sum of the sensible heat and the heat required for CO₂ desorption (grey bar + red bar in Additional Fig. 1) was similar for two adsorbents (2.63 and 2.44 GJ tCO₂⁻¹ for PEI/SiO₂ and EB-PEI/SiO₂, respectively). This is because the benefit of the large CO₂ working capacity of PEI/SiO₂ was cancelled out by the excessively strong adsorption of CO₂. In contrast, the disadvantage of the relatively small CO₂ working capacity of EB-PEI/SiO₂ was compensated by the mild adsorption of CO₂. In result, the heat required for H₂O desorption (blue bar) played the most critical role in

determining the overall heat requirement for adsorbent regeneration. The EB-PEI/SiO₂ showed much smaller H₂O adsorption than PEI/SiO₂ because of increased hydrophobicity. This led to a quite low overall heat requirement (2.66 GJ tCO₂⁻¹), even without assuming any possibility for heat exchange. In our ongoing work, we have designed various amine polymers of different structures and carried out similar heat estimations. The results indicate that the most important factor for reducing the overall heat requirement is the suppression of H₂O adsorption rather than increasing the CO₂ working capacity. These results clearly reveal that, counterintuitively, high CO₂ working capacity is not the most decisive factor in determining the overall energy consumption for CO₂ capture.

Point 2: The large difference between the actual uptake before and after functionalization (4.05 vs. 1.84 mmol/g; page 2) was dismissed in favour of more comparable “working capacities” using pure CO₂ at high adsorption temperature (110 C) to remove CO₂ adsorbed at lower temperature (1.98 vs. 1.62 mmol/g; 18% decrease nonetheless). I understand that using pure CO₂ as a purge gas during desorption may allow the separation of high purity carbon dioxide, but because the small difference between adsorption and desorption temperatures, and the large difference in the partial pressure of CO₂ during adsorption (0.15 atm) and desorption (1 atm), make this procedure unattractive. Actually with the exception of this work and previous work by the same group (Ref. 16), I am not aware of any other paper using this approach. The use of steam seems to be more popular.

Response 2: Even though the process is not the main topic of this study, we prepared the following answers for the reviewer`s question. Because of the significant electrical energy required for compressing or evacuating a large volume of flue gas, temperature swing adsorption (TSA) has been proposed as a more suitable process than pressure swing adsorption (PSA) or vacuum swing adsorption (VSA). For obtaining high-purity CO₂ suitable for further compression (>95%), the simplest way to desorb CO₂ from the saturated adsorbents is to use purified CO₂ as a sweeping gas at elevated temperatures. In earlier studies [for example, Veneman, R. *et al. Chem. Eng. J.* **207–208**, 18–26 (2012); Ntiamoah, A. *et al. Ind. Eng. Chem. Res.* **55**, 703–713 (2016)], TSA conditions similar to those employed in this study (using CO₂ as a stripping gas) were used for the demonstration of continuous CO₂ capture systems. In the cases of many amine-containing adsorbents, steam and/or vacuum were often combined with TSA, because of the limited regenerability of typical amines (with excessively strong CO₂ adsorption) and low chemical stability (*e.g.*, urea formation). The use of steam and/or vacuum can decrease the CO₂ partial pressure, which can substantially decrease the adsorbent regeneration temperature. However, these processes can be considered a modified TSA process because the major driving force for CO₂ desorption is still heat. If certain materials can show good performances in a conventional TSA process, they are also likely to show good performances in the modified TSA processes. For instance, if the EB-PEI/SiO₂ having moderate affinity for CO₂ is used

for steam-assisted TSA, we can use less steam or can further decrease the regeneration temperature compared with the PEI/SiO₂.

In addition, the benefits of these modified TSA processes over conventional TSA need to be carefully examined long-term at a large scale. In the case of steam-assisted TSA, the use of high-concentration steam demands high hydrothermal stability of the adsorbents and can also cause severe amine leaching. The use of excessive steam can increase heat consumption and requires an additional water management system. In the case of vacuum application, electricity consumption could be substantial, and it may be difficult to find a pumping system with a sufficiently large capacity suitable for treating a large volume of flue gas. If amine-containing adsorbents can be efficiently regenerated mainly via thermal driving force (as in the case of EB-PEI/SiO₂), there is no particular reason to use steam and/or vacuum. Indeed, the amine-grafted MOFs developed by Jeffrey R. Long (UC Berkeley) can be efficiently regenerated even under CO₂-rich atmospheres. These adsorbents were regenerated under 100% CO₂ at elevated temperatures [*Nature* **519**, 303–308 (2015); *J. Am. Chem. Soc.* **139**, 13541–13553 (2017)], similar to our case.

The present study was carried out with the support of the Korea CCS 2020 Project, which is one of the biggest research consortia in Korea (40 research institutes/universities have joined). Our experts in process engineering regarded the TSA process using a fluidized bed as the most promising process for CO₂ capture for coal power plants, based on the compactness of the capture system and the possibility to produce high-purity CO₂.

Point 3: Some of the most insightful papers on oxidative degradation of supported amines (DOI: 10.1039/c3cp53928h; DOI: 10.1002/chem.201300864, and possibly DOI: 10.1021/ef4001067) were overlooked or ignored.

Response 3: We appreciate the suggestions of very important papers. We have cited all the references recommended by the reviewer (refs. 16, 17, and 32 of the revised manuscript).

Point 4: Some important statements are not correct (see also comment 5), for example in page 2, lines 44-45, authors reported (based on literature sources) that “urea formation can be inhibited by using secondary amines rather than primary adsorbent regeneration (12,15,26,35).” I am not sure that this statement has been reported in (or can be drawn from) any the references cited. However, in reference 13 (Fig. 1), it was demonstrated that secondary amine does not form urea, but only when it is isolated. When it is not isolated, as in linear PEI, it does form cyclic urea, albeit the process is slower than in regular (branched) PEI.

Response 4: We greatly appreciate the reviewer's very thoughtful comments. We realized that we deleted part of this sentence by mistake, which resulted in a confusing sentence with incorrect reference numbers. We rewrote this part as follows: "For instance, urea formation can be inhibited by selectively using secondary amines rather than primary amines^{24,34} or injecting steam during adsorbent regeneration^{14,22,23,28}." For refs. 24 and 34, we have cited the papers [Sayari, A. *et al. J. Am. Chem. Soc.* **134**, 13834–13842 (2012); Sayari, A. *et al. Langmuir* **28**, 4241–4247 (2012)], respectively. For refs. 14, 22, 23, and 28, we have cited [Li, W. *et al. ChemSusChem* **3**, 899–903 (2010); Heydari-Gorji, A. *et al. Ind. Eng. Chem. Res.* **51**, 6887–6894 (2012); Sayari, A. *et al. J. Am. Chem. Soc.* **132**, 6312–6314 (2010); Hammache, S. *et al. Energy Fuels* **27**, 6899–6905 (2013)], respectively.

Point 5: Lines 101 to 105 "Notably, other studies reported that secondary amines are generally less stable than primary amines (29,30). We believe that the increased oxidative stability of EB-PEI/SiO₂, despite the increased secondary amine contents, can be attributed to the generation of abundant hydroxy (-OH) groups after EB-functionalization, which can form hydrogen bonding with nearby amines". It is correct that Ref. 29 and 30 reported that grafted propylamine (primary) is more air-resistant than grafted N-methyl propylamine (secondary). Nonetheless, this comparison is valid only for isolated amines. In polyethylenimines, the linear polymer (all secondary amines) seems to be more resistant than the branched polymer (mixture of different amines), see DOI: 10.1039/c3cp53928h. Therefore, the increased resistance may be associated with the elimination of primary amines, and not necessarily associated with OH groups.

Response 5: We appreciate the reviewer's thoughtful reinterpretation, which we strongly agree with. We rewrote the paragraph as follows: "As Sayari pointed out, the oxidative degradation of amines in the PEI-type polymers is significantly affected by the co-existence of different types of amines¹⁶. Therefore, the increased stability of EB-PEI/SiO₂ might originate from the increased portion of secondary amines at the expense of primary amines. Alternatively, it can also be attributed to the generation of abundant hydroxy (-OH) groups after EB-functionalization, which can form hydrogen bonds with nearby amines. Chuang *et al.* reported that oxidative stability of amines could be improved in the presence of additives containing hydroxy groups (*e.g.*, polyethylene glycol) because of their abilities to form hydrogen bonds with amines¹⁵."

We also significantly rewrote the introduction (lines 52–59, page 3) to differentiate the cases of isolated amines and polymeric amines: "The oxidation rate significantly depends on the amine structures. The isolated primary amines are known to be more stable than the isolated secondary amines^{30,31}. In the case of PEI, the linear PEI mainly containing secondary amines is more stable than the branched PEI with a mixture of primary, secondary, and tertiary amines¹⁶. The results indicated that the co-existence of different types of amines can affect the oxidative degradation of amines. The

polymers with only distant primary amines such as poly(allyamine) are more stable than conventional PEI¹⁷. Recently, the use of propylene spacers between amine groups has been found to substantially increase amine stability compared with the PEI containing ethylene spacers¹⁸.”

Point 6: Supplementary Table 1. While the weight of PEI increased by 62% upon reaction with epoxybutane to obtain EB-PEI, the iron and copper content remained constant in both polymers (see also page 5, line 114). This is unrealistic. As for the other metals, although their concentrations are much smaller, their variations from PEI to EB-PEI are questionable.

Response 6: Trace metal impurities are present not only in the PEI, but also in the 1,2-epoxybutane (Fe: 2.6 ppm, Cu: 1.2 ppm) and in the methanol solvent (Fe: 1.3 ppm, Cu: 0.3 ppm) used for the functionalization. There is also the possibility of metal leaching from labware during the synthesis of EB-PEI. Therefore, such simple calculation may not be appropriate. Indeed, when we did not pay attention on metal contamination (*e.g.*, careless use of a stainless-steel spoon or a container during the synthesis of EB-PEI), even higher metal contents could be observed in EB-PEI than in PEI.

Point 7: The main problem of oxidative degradation may actually not be associated with the small amount of oxygen in the feed, since CO₂ has an inhibiting effect on oxidative degradation, and the adsorption is usually carried out at low temperature. The problem may be the cooling step (after desorption of CO₂) since the cheapest way would be to cool with air. That is another problem.

Response 7: For the aeration of regenerated adsorbents in the fluidized bed operation, our process engineers plan to use flue gas instead of air, especially in the hot reactor region. This practice can help extend the lifetime of the adsorbent and increase the purity of separated CO₂.

Reviewer #2

This well-written manuscript presents studies that continues the authors' previous work in creating oxidatively-stable amine sorbents for CO₂ capture. Previously, the authors' work functionalized poly(ethylenimine) (PEI, a common aminopolymer used for CO₂ capture) with hydroxybutyl groups to improve the oxidative stability (ref. 17). This manuscript presents a simple strategy for modification of the oxide support to further improve the oxidative stability of these materials; this strategy is expected to be largely applicable to a variety of aminopolymer-based CO₂ sorbents and will be of interest to those working in the field. The manuscript should be published essentially as is, with only a couple of minor comments that the authors could consider.

The authors introduce the use of metal-chelating groups on the oxide support to remove trace amounts of metal impurity that come with commercially-available PEI. This strategy is combined with their previous PEI-functionalization strategy to produce a solid CO₂ sorbent with high CO₂ capacity and stability against oxidative degradation even after exposure to aggressive oxidizing environments for long time periods. Adsorption is performed from a simulated flue gas stream, and regeneration is performed in a CO₂ stream, thus generating a purified CO₂ product. The authors characterize the first-order degradation kinetics of their sorbents with the different treatments, demonstrating that the combination of PEI functionalization and introduction of metal-chelating groups has an additive effect on improving the oxidative stability. The work is new in demonstrating that a simple and inexpensive modification to the solid oxide support, in addition to modifying the polymer that goes in that support, can have dramatic consequences in improving the long-term viability of solid sorbent materials for CO₂ capture. The data are presented clearly, with the appropriate comparisons and control experiments. The data do not appear to have any statistical treatment, and it is unclear whether experiments have been repeated. However, the trends in stability are clear as the loading of the metal chelators is increased, suggesting that the conclusions are qualitatively valid.

Point 8: It would be helpful if the authors could present a discussion of how to distinguish between the two mechanisms of CO₂ capacity loss presented on page 7 (amine oxidation into amide/imine, and loss of amines by fragmentation). It is currently unclear how much of the capacity loss should be attributed to loss of amine content in the material, and how much to oxidation, though both appear to be suppressed by addition of the metal-chelating group. Additionally, it would be helpful if the authors could discuss how introduction of these metal chelators helps to suppress chain fragmentation.

Response 8: The chain degradation itself might be the consequence of complex amine oxidation reactions. Therefore, metal chelators could also suppress chain fragmentation. We found an important publication explaining the scission of PEI via oxidative degradation [Idris, S.A. *et al. Polymer Int.* **55**, 1040–1048 (2006), ref. 47 of the revised manuscript]. According to that paper, the ethylenediamine unit of PEI can be decomposed by oxidation into formamide and hemi-aminal, which can subsequently be decomposed into formic acid, amines, and imines. Alternatively, the imines generated at the middle of the polymer backbone can also be hydrolysed to produce amine and aldehydes, which can lead to chain cleavage. We added discussion of these processes in lines 177–181, pages 7–8 of the revised manuscript as follows: “The chain degradation may be the consequence of complex amine oxidation reactions. It has been proposed that the ethylenediamine unit of PEI can be decomposed by oxidation into formamide and hemi-aminal⁴⁷, which can subsequently be decomposed into formic acid, amines, and imines. Alternatively, the imines generated at the middle of the polymer backbone can be hydrolysed to produce amine and aldehydes, which can also result in chain cleavage.”

Reviewer #3

This paper reports that the amine-containing CO₂ adsorbents prepared via combining two strategies (functionalization of PEI with 1,2-epoxybutane and the poisoning of metal impurities with chelators) show the significantly enhanced oxidation stability even after 30 days-long aging in O₂-containing flue gas at 110 °C. I found this article certainly scientifically interesting, making me recommend its publication in *Nature Communications*. However, I have a couple of comments to be properly answered prior to its publication (see below).

Point 9: The authors carried out the CO₂ adsorption-desorption experiments under practical temperature swing adsorption (TSA) conditions. CO₂ adsorption was performed using a wet flue gas containing 15% CO₂, 10% H₂O, and N₂ balance at 60 °C for 30 min and desorption was performed under 100% CO₂ at 110 °C for 30 min. I wonder how the authors treated the gas during decreasing temperature from 110 to 60 °C before moving to the next CO₂ adsorption step. This is very important because the remaining CO₂ gas or other carrier gas (i.e., He or Ar) can somehow influence the CO₂ working capacity of the solid adsorbents. In the case of zeolites X and A, the adsorbents can be partly regenerated under He or Ar flow even at atmospheric condition (25 °C and 1 bar). The author should clearly mention the whole experiment conditions.

Response 9: In the present work, we actually did not carry out the cyclic experiments, but instead compared the single-cycle CO₂ adsorption-desorption profiles before and after isothermal aging under O₂-containing atmospheres. The procedure is given in the Methods section. Prior to measurements, all fresh or aged samples were degassed at 100 °C for 1 h under N₂ flow (50 cm³ min⁻¹). CO₂ adsorption was carried out using a simulated wet flue gas containing 15% CO₂, 10% H₂O, and N₂ balance at 60 °C. After 30 min adsorption, the gas was switched to 100% CO₂ flow (50 cm³ min⁻¹) and the temperature was increased to 110 °C (ramp: 20 °C min⁻¹). In our earlier paper showing the data of fifty TSA cycles [*Nat. Commun.* **7**, 12640 (2016)], we repeated the adsorption-desorption protocol except for the pre-degassing step. Before decreasing the temperature back to the adsorption temperature for the next cycle, we switched the gas from 100% CO₂ to a simulated flue gas containing 15% CO₂. CO₂ adsorption can take place even during cooling, and adsorption equilibrium is finally achieved at the adsorption temperature. If we repeat the adsorption-desorption cycles, we can get CO₂ adsorption-desorption 'swings' and the swing amplitude becomes the CO₂ working capacity. Therefore, in our previous cyclic measurements using TGA-MS, we did not purge any noble gases between cycles. Our previous experimental conditions for cyclic experiments are similar to those used by Jeffrey R. Long and colleagues [*Nature* **519**, 303 (2015)].

Point 10: In their previous study, the author stated that the amine-functionalized porous material may suffer from serious amine deactivation due to urea formation under the desorption conditions at temperatures higher than 120 °C. They indeed regenerated the samples under at 130 (Energy Environ. Sci., 2016, 9, 1803-1811) or 120 °C (Nat. Commun., 2016, 7), and successfully overcame the adsorbent deactivation. However, in the submitted ms, the adsorbents were aged at 110 °C only, which is lower by 10 °C than the amine deactivation temperature (120 °C) and the same as the pretreatment temperature (110 °C). I would like to know the reason the authors selected 110 °C as an ageing temperature.

Response 10: Similar to other researchers, we also believed that achieving high CO₂ working capacity is important in the design of CO₂ capture processes. Therefore, to achieve ~100% desorption of CO₂, we used a higher desorption temperature (120 °C for EB-PEI/SiO₂). However, as explained in Response 1, we recently realized that increased CO₂ working capacity does not significantly contribute to the reduction of heat requirement for the CO₂ capture process. In the case of EB-PEI/SiO₂ with moderate affinity for CO₂, almost 90% CO₂ desorption can still be achieved even at the mild regeneration temperature of 110 °C. The slightly decreased CO₂ working capacity does not increase the sensible heat (the ΔT reduced by 10 °C actually contributes to a slight reduction of sensible heat). If we decrease the regeneration temperature by 10 °C, the amine degradation pathways (*i.e.*, urea formation and oxidative degradation) can be significantly suppressed because these are also ‘chemical reactions’. Because the adsorbent should be stable over several months in practical applications, the mild desorption temperature of 110 °C seems much more beneficial. In subsequent publications, we will clarify this point with energy calculation data. We are still learning in this project.

REVIEWERS' COMMENTS:

Reviewer #2 (Remarks to the Author):

The authors have prepared a thoughtful response to the comments from all of the referees. In particular, the discussion surrounding the distinction between oxidative stability of isolated amines vs. polymeric amines is well thought out and is a nice summary of the literature. Therefore, I recommend publication of this manuscript in Nature Communications.

Reviewer #3 (Remarks to the Author):

Reviewer's Report for Nature Communications.

Author(s): Kyungmin Min, Woosung Choi, Chaehoon Kim, and Minkee Choi

Manuscript Number: NCOMMS-17-28665A

Manuscript Type: Article

Comments:

This contribution reports on significantly enhanced oxidation stability on amine sorbents for CO₂ capture. I have carefully checked the resubmitted manuscript and found that it has been properly revised based on my comments. So I think this ms is publishable as the present form in Nature Communications.